# L2BGAN: AN IMAGE ENHANCEMENT MODEL FOR IMAGE QUALITY IMPROVEMENT AND IMAGE ANALYSIS TASKS WITHOUT PAIRED SUPERVISION

## ABSTRACT

The paper presents an image enhancement model, L2BGAN, to translate low light images to bright images without a paired supervision. We introduce the use of geometric and lighting consistency along with a contextual loss criterion. These when combined with multiscale color, texture and edge discriminators prove to provide competitive results. We perform extensive experiments on benchmark datasets to compare our results visually as well as objectively. We observe the performance of L2BGAN on real time driving datasets which are subject to motion blur, noise and other artifacts. We further demonstrate the application of image understanding tasks on our enhanced images using DarkFace and ExDark datasets.

## 1 INTRODUCTION

Image enhancement is a prerequisite for many computer vision based image understanding tasks. In particular, it is very crucial to enhance low light or dark images to obtain images which not only have better image aesthetic quality but can also be suitably processed for object detection and face detection tasks. The task by itself is one of the earliest studied domains in computer vision, but the continuous evolution of computational resources and deep learning architectures has generated a paradigm shift towards the latter approach. A thorough literature review shows three main ways in which the problem is approached: histogram equalization (Ibrahim & Pik Kong, 2007; Nakai et al., 2013), Retinex-based Guo et al. (2017); Wang et al. (2016); Fu et al. (2016), and machine learning based (Lv et al., 2018; Wang et al., 2019; Jiang et al., 2021; Guo et al., 2020). The primary challenge while dealing with low light image enhancement tasks is that there are many noise sources in the acquisition of poorly lit scenes. These include readout, photon shot, dark current, and fixed pattern noises, in addition to photon response non-uniformities. The noise level increases while treating lightness and contrast of low light images, more so in case of compressed-dynamics images. Applying a denoising filter prior to light enhancement will result in blurring, while the reverse causes noise amplification as seen in Fig. 1. Hence, dealing with denoising and low-light enhancement problems simultaneously using a learning based approach seems to be the optimal choice.

The success of deep learning enhancement models depend on the availability of large scale anno-

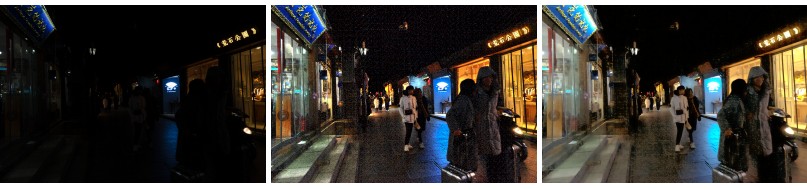

Figure 1: Original Dark Image, Processed Image with LIME (noise amplified), Processed Image with MBLLEN (artifacts due to smoothing), from left to right

tated data. For the present problem, there is a requirement of paired images such that the low-light image serves as input while its brighter counterpart serves as the target image. The Adobe 5k dataset (Vladimir et al., 2011) serves as a benchmark used by many researchers for this task. This dataset provides an original low light image, retouched by five different photographic experts, one of which

is used as the target of supervised training. Some researchers have also used a synthetic paired set for the same by intentionally transforming a bright image to generate its dark counterpart. This transformation being global, its suitability for challenging real time images is arguable. Ideally, a training procedure without the requirement of paired supervision is the most appropriate solution. Very few works have been reported on low light image enhancement using unpaired images (Jiang et al., 2021; Ni et al., 2020). Moreover, even though image to image style translation without paired supervision is a popular approach Anoosheh et al. (2019); Fu et al. (2019); Zhu et al. (2017b), not many have considered the mapping of low light images to bright light images as a style transfer problem. In this work, we consider the low light image enhancement task as an image style transfer problem. We train a deep neural architecture using unsupervised image pairs. This promotes the use of available real time dark and bright images without the need of pairwise annotations or synthetic treatment. We resort to GANS for training using two encoders and two decoders. We use three discriminators to learn color, texture, and edges separately. The unsupervised image to image translation problem is challenging as it aims to recover a joint probability given a marginal probability. For this purpose, one of the most common approaches is to add a cycle consistency constraint along with the adversarial losses. This bijective constraint becomes restrictive in cases as the task at hand, where the low light image domain may contain less information compared to its bright counterpart. To handle this situation we use a geometric consistency constraint which ensures that an image from one domain and its geometric transformed version generate the same image in the other domain. Additionally, a contextual loss constraint ensures that the context and semantics of the images are preserved. This paper has the following main contributions: (i) it poses the low to bright light enhancement problem as an unpaired image translation problem; (ii) it uses a geometric and an illumination consistency constraint in the images; (iii) it uses a contextual loss to measure semantic similarity; (iv) it uses multiscale color, texture and edge discriminators per domain; (v) it demonstrates the effectiveness of the approach for image understanding tasks. While none of such items are individually novel, exploiting their ensemble effect in this task is indeed a significant novel contribution. Our paper is organized in the following manner. We provide a related background of the task in Sec. 2. The proposed technique is discussed in details in Sec. 3. Experimental results and conclusion is provided in Secs. 4 and 5 respectively.

## 2 RELATED BACKGROUND

**Histogram based enhancement** aims to improve the image quality by modifying the image histogram. This can result in overstretched contrast which lacks in naturalness. Using this framework different techniques have been developed. BPDHE (Ibrahim & Pik Kong, 2007) uses an HE on a dynamic range expanded version of sub-histograms obtained by the local maxima of the input image histogram. FHSABP (Wang et al., 2008) solves a convex optimization problem to obtain the flattest target histogram with the brightness preservation constraint. Due to their brightness preservation, (Ibrahim & Pik Kong, 2007) and (Wang et al., 2008) are not suitable to tackle the low light image enhancement, but prevent overstretching. Morover, (Ibrahim & Pik Kong, 2007) and (Wang et al., 2008) do not consider relationships between adjacent pixels, whereas methods such as HMF (Arici et al., 2009), CEBGA (Hashemi et al., 2009) and DHECI (Nakai et al., 2013) do. In HMF (Arici et al., 2009) the target histogram is obtained via a parametric optimization problem, involving the local variance of pixels. In CEBGA(Hashemi et al., 2009) a genetic algorithm is adopted to enhance the contrast of the images modifying the histogram, using as a fitness function the number of edges in the enhanced image. DHECI (Nakai et al., 2013) performs a histogram equalization on the differential intensity histogram and the differential saturation histogram from the HSI color space. More advanced techniques which embed contextual information are CVC (Celik & Tjahjadi, 2011) and LDR (Lee et al., 2013), which use 2D histograms.

**Retinex based methods** use the assumption that an image can be pixel-wise decomposed into reflectance and illumination. In (Li et al., 2011) a classic approach is proposed, where the lightness is first decomposed in reflex lightness and ambient illumination. Reflectance is then extracted from reflex lightness, while the ambient illumination is log-transformed and used for the output image. A widely used framework for the Retinex based enhancement methods is the fusion based framework, as in FEMWII Fu et al. (2016) and FMSBIE Wang et al. (2016). Both estimate the illumination using the pixel-wise maximum in the RGB color space, which is used to obtain the reflectance. Three different modified illuminations are then obtained through different techniques to improve bright-

ness and contrast. These illumination maps are then fused with a multiscale approach. Multiscale techniques are also used to extract reflectance (Liu et al., 2016)

Another approach for the illumination estimation is proposed in LIME (Guo et al., 2017), where an optimization problem is used to obtain a smooth, structure-preserving illumination map, which is then enhanced through a gamma correction. As the noise is critical in poorly illuminated images in (Guo et al., 2017) a denoising techinque is also proposed: the output reflectance is the linear combination of the original one and a BM3D-filtered one, using the illumination as the weight to prevent the oversmoothing of bright areas. In (Li et al., 2011), Fu et al. (2016), Wang et al. (2016), (Liu et al., 2016), and (Guo et al., 2017) reflectance is obtained by the estimated illumination. More advanced techniques, such as SRIE Fu et al. (2016) and JIEPMR Cai et al. (2017), jointly decompose reflectance and illumination. While the Retinex theory has proved to be suited for low light image enhancement, this approach suffers from color distortion and hand-crafted illumination manipulation. Learning based methods can overcome these problems, while also providing better denoising, which is crucial in low light images.

**Data learning methods** include a variety of methods from deep learning, such as autoencoders, Deep CNNs and GANs, all of which have proved to be suitable techniques to enhance image quality and brightness.

In LLNet (Lore et al., 2017), GLADNet (Wang et al., 2018) and LLED-Net (Li et al., 2020), denoising *autoencoders* are used. Despite the good results, the simplicity of these methods limits their capability compared to more complex learning methods. In (Ren et al., 2019) multiple autoencoders are used to work separately on the content and on the edges of the image while also using CNN, RNN and a more advanced loss function, including a discriminator and a VGG-16 net. An autoencoder is also used for DeepUPE (Wang et al., 2019) to learn a image-to-illumination mapping used for a Retinex-fashioned enhancing, while considering reconstruction loss, color loss and smoothness of the illuminance. In (Jiang & Zheng, 2019), an end-to-end U-Net is used, with a 3D convolution in order to deal with videos, preventing flickering in consecutive frames, and using the GRBG components from the camera sensor as an input. In MBLLEN (Lv et al., 2018), a CNN is applied to the input image. Then, the output of each layer is fed to independent autoencoders. The obtained output image is a weighted sum with learnable weights of the autoencoders output. Considering strutural, contextual and regional loss for the training. While autoencoders and U-Net are convenient for image-to-image nets, methods such as Tao et al. (2017) and (Ignatov et al., 2017) use only deep neural networks. In particular, these use residual *convolutional neural networks* to mitigate the vanishing gradient problem in deep networks. In (Ignatov et al., 2017), the CNN enhances images from low-end devices to professional DSLR quality, using a dataset with multiple acquisition of the same scene from different devices for the training, which is difficult to obtain. Tao et al. (2017) uses special-designed residual CNN modules to create a deep CNN, but is then trained using only the SSIM. Apart from (Wang et al., 2019), also RetinexNet (Wei et al., 2018) and Kind++ (Zhang et al., 2021) augment the Retinex theory, outperforming classic Retinex based methods. Both (Wei et al., 2018) and (Zhang et al., 2021) use a CNN to decompose the reflectance and illumination components. In both cases, denoising take advantage of the illumination map, as in (Guo et al., 2017), to improve the result over plain denoising. In (Wei et al., 2018), the BM3D is used for denoising whereas in (Zhang et al., 2021) a specific CNN with multi-scale illumination consideration is proposed. For illumination adjustment (Wei et al., 2018) uses an encoder decoder structure while (Zhang et al., 2021) adopt a CNN with a free parameter to modify the adjustment ratio. A common critical aspect about the methods considered above is *data availability*. Since these methods are supervised, they require paired low/normal light images of the same scene. A widely used way to obtain them is to artificially darken normal light images, but this impacts the naturalness of the results. Recently, methods such as (Wei et al., 2018) and (Zhang et al., 2021) used datasets with images of the same scenes with different exposures. This proved to be effective, but such datasets are scarce and tedious to create. To overcome the data availability issue, several unsupervised and unpaired supervised methods have been developed. ZDCE (Guo et al., 2020) is an unsupervised low-light image enhancer which uses a CNN to obtain pixel-wise enhancement curves to adjust the input image, using non-reference loss functions for the training. EGAN (Jiang et al., 2021) and UE-GAN (Ni et al., 2020) use a GAN to obtain a light enhancement without paired supervision, using a decoder-encoder for the generation. In (Jiang et al., 2021) a local and a global discriminator are used while in (Ni et al., 2020) only one multi-scale discriminator is used. The promising results of (Jiang et al., 2021) and (Ni et al., 2020), together with the availability of unpaired low/normal light

images, makes GANs an appealing approach for low light image enhancement techniques.

Our contribution is similar to EGAN and UEGAN as it uses a GAN architecture with unpaired supervision. Different from these, we do not use a global attention. We rather use separate discriminators for color, texture and edge. Further we make use of cycle consistency, geometric and illumination consistency with contextual loss instead of perceptual losses.

## 3 METHODOLOGY

CycleGAN(Zhu et al., 2017a) was one of the first technique used for unsupervised image translation, where a cycle consistency loss was used to learn the semantic dissimilarity between the two transfer domains. This replaced the direct paired loss. This translation is an open ended problem, as a single image in a domain may result in multiple images in the other domain. While there are techniques which deal with the multimodal nature of the problem (Zhu et al., 2017b) by random sampling or using more than one target, we focus on generating a single translation at a time. Other than the cycle consistency, many regularizations are used on the generators to enable the creation of real-generated images while using unpaired learning. These include penalizing the distance in the latent space, perceptual loss, and forcing the generators to be close to the identity function. In this work we relax the number of regularizations on the generator while ensuring that the generator is able to learn inverse mappings. We aim to map images in a low light domain, $L_x$, to bright images $B_y$. Both $L_x$ and $B_y$ consist of a finite number of samples. An image $X_{real}$ belonging to $L_x$ can be mapped to $Y_{fake}$ in domain $B_y$ using an encoder $E_x$ and a decoder $D_y$, $Y_{fake} \rightarrow D_y(E_x(X_{real}))$. Similarly, $X_{fake} \rightarrow D_x(E_y(Y_{real}))$. The cycle consistency can be computed using $Y_{fake}$ to regenerate $X_{recon}$ using $D_x(E_y(Y_{fake}))$. Additionally, to set up the geometric and lighting consistency constraint, we transform $X$ into $X\_g$ and $X\_l$, where $X\_g$ is a 90 degrees rotation of $X$ and $X\_l$ is a gamma transformation of $X$. When $Y_{fake}$ is generated, it should be similar to $Y\_l_{fake}$ and $Y\_g_{fake}^{-1}$. The inverse mapping refers to the inverse rotation of the reconstructed image. This ensures that the generator does not add further artifacts in the image while transforming it. Two discriminators $Dc^L$ and $Dc^B$ are used for the two domains. Further, each discriminator is divided into three parts: $Dc_{xc}$, $Dc_{xt}$ and $Dc_{xe}$. These aim at discriminating color ($xc$), texture ($xt$) and edge ($xe$) differences between the real and the generated images in $L_x$ domain. The inclusion of these discriminators facilitates learning color distributions, texture distributions and edge distributions from unpaired images. The color discriminator uses blurred RGB images. More precisely, since we have observed increased performances using multiscale discriminators, we use four discriminators: $Dc_{xc1}$, $Dc_{xc2}$, $Dc_{xt}$ and $Dc_{xe}$ instead of three. Here $Dc_{xc1}$ and $Dc_{xc2}$ denote RGB images blurred with two different factors. A grayscale version of the image is used as the texture image, while the edge image is obtained via a Prewitt operator. The overall objective function is shown in Eq. 4. It includes the general adversarial loss $L_{gan}$ along with the cycle reconstruction loss $L_{cyc}$ and the consistency loss $L_{cyc\_con}$. While the adversarial loss and the cycle reconstruction loss are computed over both domains, the consistency losses are computed only on the target domain. Further, instead of using an $L_1$ loss a contextual loss is used in case of geometric and lighting consistency measurements. There are significant differences in the network performance with variations in the loss functions. We split the training process into four different stages: (i) first, we train the network with adversarial loss $L_{gan}(X,Y)$ and cycle consistency loss $L_{cyc}(X, X_{recon}, Y, Y_{recon})$ using color, texture and edge discriminators. We call this version L2B_base; (ii) we next use geometric consistency $L_{cyc}(Y_{fake}, Y\_g_{fake}^{-1}, X_{fake}, X\_g_{fake}^{-1})$ along with adversarial loss and cycle-consistency. We use only color images in the discriminator and use $L_1$ loss criterion for consistencies. We denote this as GC; (iii) in the third phase we replace $L_1$ loss criterion in (ii) with contextual loss. We name the result GC_con; (iv) to obtain the final network version we add $L_{cyc\_con}$ loss to L2B_base. Unlike GC we only compute the consistencies in the target domain. Also, the $L_1$ loss criterion for consistency measurements is replaced by contextual loss. We call the final result L2BGAN. The same is depicted in figure2

$$L_{gan}(X,Y) = \mathbb{E}_b[(Dc(Y)-1)^2] + \mathbb{E}_a[(Dc(D_y(E_x(X))))^2] \qquad (1)$$

$$L_{cyc}(X, X_{recon}, Y, Y_{recon}) = \mathbb{E}_a[||X_{recon} - X||_1] + \mathbb{E}_b[||Y_{recon} - Y||_1] \qquad (2)$$

$$L_{cyc\_con}(X, X_{recon}, Y, Y_{recon}) = \mathbb{E}_a[||X_{recon} - X||_c] + \mathbb{E}_b[||Y_{recon} - Y||_c] \qquad (3)$$

$$L_{tot} = L_{gan}(X,Y) + L_{gan}(Y,X) + L_{cyc}(X, X_{recon}, Y, Y_{recon})$$
$$+ L_{cyc\_con}(Y_{fake}, Y\_g_{fake}^{-1}, Y_{fake}, Y\_l_{fake}) \qquad (4)$$

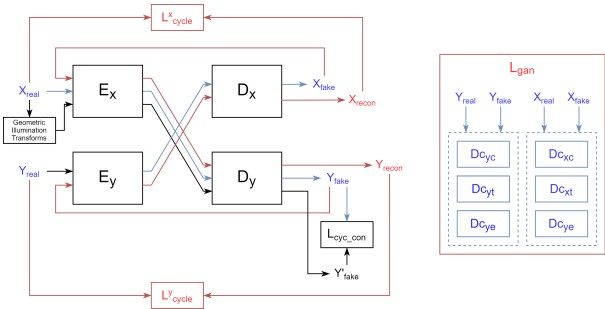

Figure 2: Block Diagram of L2BGAN.

# 4    EXPERIMENTAL RESULTS

We used a mixture of images from two different datasets for training our network. We included images from the Adobe Dataset and the BrighteningTrain dataset. 3000 images were randomly selected for training. The network was trained for a total of 150 epochs with a learning rate of 0.0002. The images were randomly cropped at size $256 \times 256$. Training was done on a workstation with an RTX6000 24GB GPU. Each training took approximately 36 hours. Initially we show some ablation study in section 4.1. For testing we use benchmark datasets DICM, LIME, MEF, NPE and backlit images. We compare the proposed work with state-of-the-art techniques like EGAN, MBLLEN, ZDCE, DeepUPE, LIME, FMSBIE. For all these methods softwares provided by the authors were utilised. Additionally we present image understanding results on the Berkeley Driving datasets and challenging low light datasets like DarkFace and ExDark. In Sec. 4.2 we present visual and objective evaluation of our technique. We use no-reference image quality assessment tools such as NIQE, PIQE and BRISQUE for objective evaluations, since ground truth for these data is not available. In Sec. 4.3 we evaluate the LOL dataset using PSNR and SSIM scores. In Sec. 4.4 we present visual and objective evaluation on the real time Berkeley Driving dataset. Finally in Sec. 4.5 we provide some evaluations of our technique on the DarkFace and Exdark datasets.

## 4.1    ABLATION STUDY

In Fig. 3 we demonstrate the performance of each network discussed in 3. We observe that the processed images have visible blocking artifacts and color saturation for (ii) and (iii). Hence, use of (a) color, texture and edge discriminator, (b) geometric and lighting consistency on target domain, (c) contextual loss in place of $L_1$ loss prove to be significant parts of the L2BGAN. These results are further backed by processing the Exdark dataset with each of the networks and computing the IQA scores for them. We observe that the best PIQE, NIQE and BRISQUE scores are obtained for $L2BGAN$ (32.24; 2.96; 22.7). The $GC$ version provides the second best scores (32.84; 3.18; 22.97). Both $L2B\_base$ and $GC\_con$ show comparatively poor performance in case of visual results as well as objective scores. For example, $L2B\_base$ scores are (36.8477; 3.5547; 25.1551).

## 4.2    EVALUATION ON BENCHMARK LOW LIGHT DATASET-A

In Table 1 we demonstrate the image quality scores obtained after preprocessing the original dark images in each case with the respective techniques. The scores shown are the average score of the entire dataset. It is observed that our proposed technique L2BGAN obtains competitive results. NIQE tool provides the best results for DCIM, LIME and MEF datasets when L2BGAN is applied. For NPE, L2BGAN provides the second best score. The LIME and NPE datasets also show highest performance with L2BGAN when PIQE image quality tool is used. It is however noted that L2BGAN does not perform well on backlit images. Also for BRISQUE tool, DeepUPE provides the maximum scores whereas L2BGAN obtains the second best scores. We also show some visual comparisons in Fig. 4. More comparisons can be found in 9. We observe that for EGAN the front

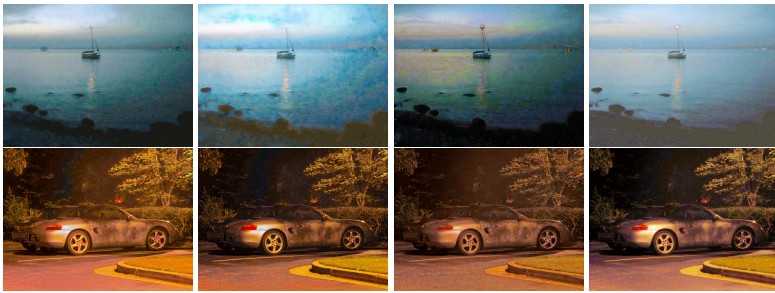

Figure 3: The images in each row from left to right are resultant images from GC, GC_con, L2B_base and the complete L2BGAN, respectively

Table 1: Objective no-reference image quality scores on different benchmark datasets. (Values in red and blue are best and second best, respectively)

| DATASET | PIQE | NIQE | BRISQUE | PIQE | NIQE | BRISQUE | PIQE | NIQE | BRISQUE |
|---|---|---|---|---|---|---|---|---|---|
| | DeepUPE | | | EGAN | | | LIME | | |
| DICM | 29.7 | 3.14 | **20.84** | **28.04** | 2.73 | 21.88 | 34.27 | 2.91 | 24.72 |
| LIME | 35.19 | 3.76 | 26.23 | **34.75** | **3.52** | **20.87** | 40.76 | 4.44 | 22.09 |
| MEF | 35.3 | 3.16 | **17.86** | **32.22** | **2.89** | 23.26 | 38.76 | 3.67 | 23.36 |
| NPE | **33.2** | **3.29** | **21.5** | 33.96 | 3.34 | 27.34 | 38.87 | 3.9 | **27.1** |
| backlit | 26.42 | 2.97 | **19.24** | **22.8** | **2.48** | **25.44** | 27.64 | 2.83 | 34.93 |
| | MBLLEN | | | ZDCE | | | FMSBIE | | |
| DICM | 40.18 | 2.91 | 27.87 | **25.94** | 2.69 | 24.87 | 29.19 | **2.66** | 21.63 |
| LIME | 52.3 | 3.89 | 30.6 | 37.26 | 3.97 | 23.75 | 38.63 | 4.21 | 27.99 |
| MEF | 54.59 | 3.52 | 32.22 | 34.63 | 3.33 | 25.27 | 34.95 | 3.39 | 23.54 |
| NPE | 45.02 | 3.46 | 31.34 | 37.62 | 3.94 | 29.1 | 40.98 | 4.01 | 28.72 |
| backlit | 42.9 | 3.21 | 37.88 | **21.17** | **2.61** | 39.64 | 29.71 | 2.99 | 28.47 |
| | L2BGAN | | | | | | | | |
| DICM | 28.27 | **2.5** | **20.89** | | | | | | |
| LIME | **33.25** | **3.26** | **21.66** | | | | | | |
| MEF | **32.96** | **2.78** | **20.94** | | | | | | |
| NPE | **29.42** | **3.31** | 27.21 | | | | | | |
| backlit | 30.31 | 3.09 | 28.29 | | | | | | |

building is relatively too bright; the same happens for ZDCE in the tower area. A histogram analysis shows that for EGAN, ZDCE, DeepUPE and FMSBIE the red and/or green channel saturate. EGAN and LIME yield noisy halos near the bright portion of the tower; in the dark sky areas away from the objects noise is visible. MBLLEN is unable to provide sharp details in the front building. The example image shown in 4 has noise artifacts. Indeed, we have selected this image to demonstrate that our technique provides the least amount of artifacts when compared to other contemporary ones.

### 4.3 EVALUATION ON BENCHMARK LOW LIGHT DATASET-B

In Table 2 we demonstrate the image quality scores obtained on LOL dataset after preprocessing the original dark images in each case with the respective techniques. As groundtruth data is available for LOL, we also show PSNR and SSIM scores for the same in Table 2. It is observed that our proposed technique L2BGAN obtains the best scores for both cases and for all the image quality assessment tools. Sample Low light image, its ground truth, and the processed version using $L2BGAN$ are shown in fig5. It is observed that PSNR (16.53) and SSIM (0.57) for the processed images are quite low. However, the original PSNR and SSIM scores are 8.18 and 0.17 respectively (averaged on 789 test images). Although the scores of the processed images are still low, they are largely improved over the original ones as well as the ones obtained using other techniques. It should be noted that these are real-world (not synthetic) images; i.e. they represent an actual problem a user may face.

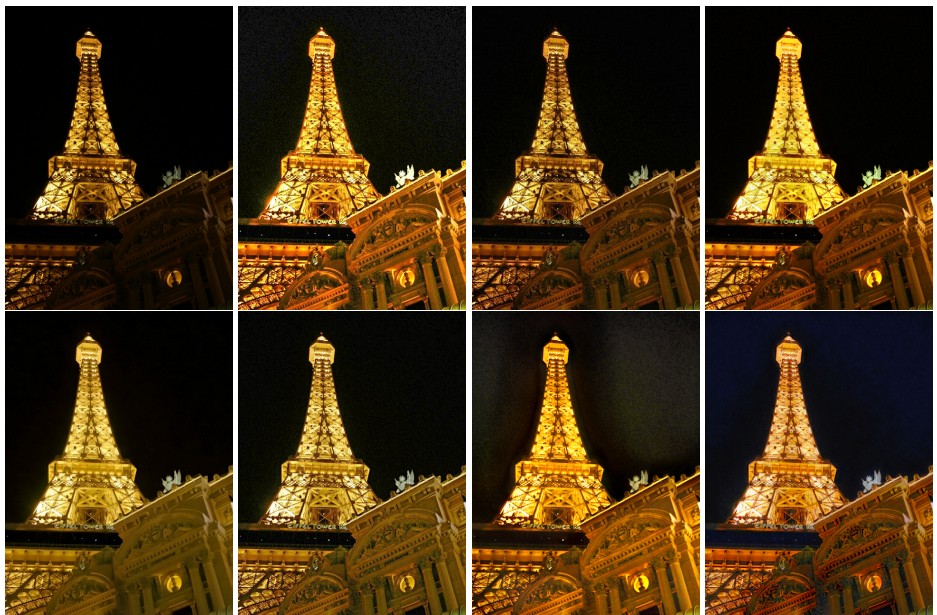

Figure 4: Sample original DICM image (top left) and its processed versions using Lime, FMSBIE, DeepUPE (first row, from second column to fourth column), and using MBLLEN, ZDCE, EGAN and L2BGAN (second row, from left to right).

Table 2: Objective no-reference image quality scores along with SSIM and PSNR on the LOL dataset

|  | DeepUPE | EGAN | FMSBIE | LIME | MBLLEN | ZDCE | L2BGAN |
|---|---|---|---|---|---|---|---|
| PIQE | 34.16 | 41.77 | 50.85 | 55.83 | 42.33 | 47.27 | **26.13** |
| NIQE | 10.63 | 7.67 | 12.04 | 11.81 | 5.41 | 11.21 | **3.15** |
| BRISQUE | 33.09 | 30.04 | 38.34 | 39.23 | 28.48 | 36.86 | **22.13** |
| PSNR | 10.70 | 15.31 | 12.16 | 14.45 | **16.59** | 13.42 | 16.53 |
| SSIM | 0.3233 | 0.4918 | 0.3995 | 0.3901 | 0.5134 | 0.4252 | **0.5742** |

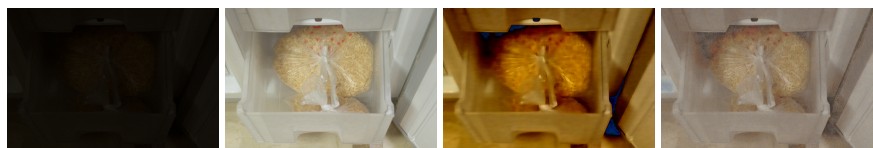

Figure 5: Original low light, GT, processed images with MBLLEN and L2BGAN from left to right

Table 3: Objective no-reference image quality scores on BDD.

|  | DeepUPE | EGAN | FMSBIE | LIME | MBLLEN | ZDCE | L2BGAN |
|---|---|---|---|---|---|---|---|
| PIQE | 67.70 | 68.75 | 72.50 | 73.65 | 64.84 | 72.01 | **61.17** |
| NIQE | 3.62 | 3.21 | 4.25 | 4.13 | 3.58 | 4.29 | **3.12** |
| BRISQUE | 46.56 | 36.53 | 46.89 | 47.85 | 46.91 | 49.02 | **36.03** |

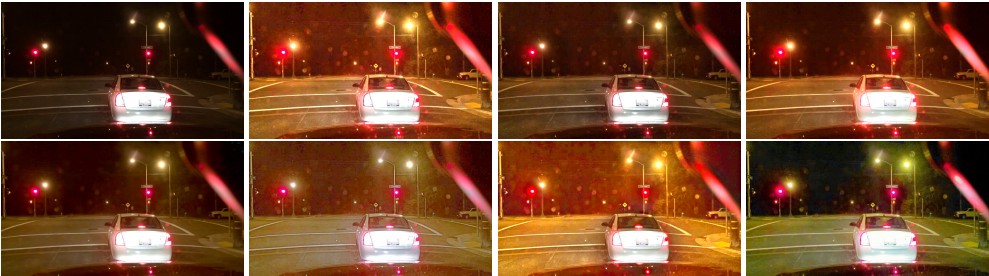

Figure 6: Sample original BDD image (top left) and its processed versions using Lime, FMSBIE, DeepUPE (first row, from second column to fourth column), and using MBLLEN, ZDCE, EGAN and L2BGAN (second row, from left to right).

## 4.4 EVALUATION ON REAL TIME DRIVING DATASET

In Table 3 we demonstrate the image quality scores obtained on the Berkeley Driving Dataset (BDD) after preprocessing the original dark images in each case with the respective techniques. The Berkeley Driving Dataset consists of road scenes captured during day and night. We select dark images by filtering images with a global mean smaller than 30. The scores shown are the average score of the entire dataset. It is observed that our proposed technique L2BGAN obtains the best scores for all the image quality assessment tools. Images shown in Fig. 6 depict that blocking artifacts and light reflections are enhanced for EGAN and LIME providing poor images. These problems are not visible for FMSBIE, MBLLEN and L2BGAN. Also, the enhancement provided by L2BGAN is smoother and brighter in the road regions without color saturation and changes in the sky region.

## 4.5 EVALUATION OF IMAGE UNDERSTANDING TASKS

One of the most important purposes of low light enhancements is to perform image understanding on it. As shown in case of Berkeley Driving dataset, a brighter and noise free image of the road will enable more accurate object detection tasks. Exdark and DarkFace are two benchmark datasets for object and face detection respectively. The DarkFace dataset has 6000 training and validation images and is particularly challenging since it presents an extremely dark environment with strong noise. Enhancing the images amplifies this noise making the face detection task very difficult. We randomly select two testsets of 500 images each from the 6000 image dataset and use the remaining 5000 as training. We obtain a mAP of 0.209 and 0.218 on the original darkface images for the two testsets when a pretrained Retinaface (trained on Widerface dataset) detector is applied. After applying a preprocessing with L2BGAN the mAP is 0.301 and 0.331 respectively. We also test the mAP after applying EGAN and obtain 0.285 and 0.331 respectively. Finetuning the pretrained network with preprocessed DarkFace images gives a mAP of 0.525 and 0.407 on L2BGAN and EGAN respectively. Fig. 7 shows some example original and preprocessed images (using L2BGAN) along with the detections obtained. More examples can be found in figures 11 and 12 in the Appendix. It is seen that the number of detected faces increases after preprocessing. We use S3FD face detectors and YOLOV3 object detectors to show visual results. Training and finetuning is obtained via Tinaface detector. Such experiments primarily aim at showing how pre-processing the images using our $L2BGAN$ model improves the results of any s-o-a detection algorithm. All experiments hence report detection results using a standard algorithm post enhancement application. For example, Fig 7 shows two false detections on the original image. On the processed image one more face is detected, with no false detections. Although it would be interesting to train the detection network jointly along with the enhancement network, it is currently not covered in our scope.

We also perform 2 simple experiments to analyze the quality of the classification features and object detection scores obtained from the L2BGAN images compared to the low light images. We take low and normal(GT) images from the LOL dataset and get L2BGAN images by processing the low images. We further compute distance $d2$ and $d1$ for each layer of VGG-19 trained with Imagenet. $d2$ and $d1$ are obtained using feature difference of normal-L2BGAN images and normal-low image pairs respectively. We then compute a distance index $(d1-d2)/(d1+d2)$ as shown in figure 8(left). We observe that (a) $d1$ is greater than $d2$, hence L2BGAN features are closer to normal for all

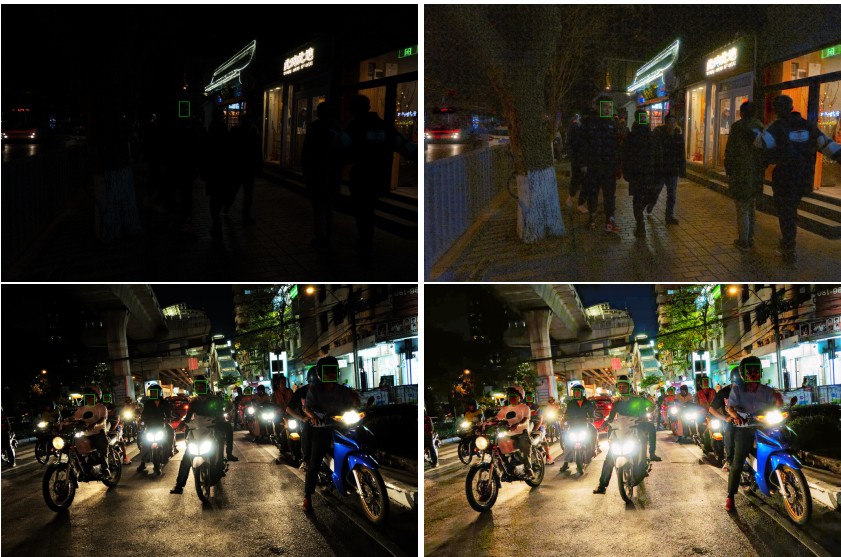

Figure 7: Face detection performances on original (first column) and preprocessed image using L2BGAN (second column) from DARKFACE (first row) and ExDark (second row) dataset.

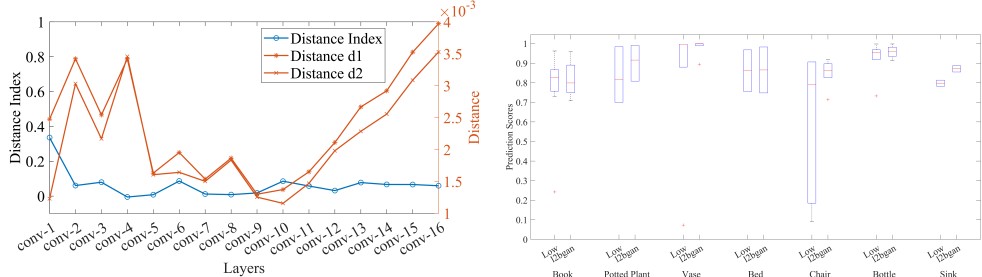

Figure 8: The figure on the left depicts the distance index obtained from classification features of low,normal and processed image. The right figure depicts the object detection scores using low and processed images.

layers,(b) although the distance index reduces as we move towards higher layers, it is still greater than 0, (c) another interesting observation is that both low and L2BGAN features are closest to the normal between conv9-10 after which they increase again. Hence extracting features from this layer may prove beneficial. In figure 8(right) we compare the object detection scores of low and L2BGAN images. 40 images are picked from the LOL dataset for this purpose. It is observed that L2BGAN in general provides a greater mean score and lower deviation.

## 5 CONCLUSION

We present an unpaired GAN based image enhancement operation using cycle consistency, geometric and illumination consistency. Visual and objective results presented on benchmark datasets show that L2BGAN provides competitive results. It is observed that L2BGAN is able to enhance real images suffering from typical artifacts, without considerably amplifying the blocking artifacts. The images show smooth enhancements without color saturation in most cases. We have particularly focused on Jpeg images even though a simple experiment on RAW data is discussed in the Appendix. Applications of L2BGAN for processing images to be used for face detection show significant improvement. It would be interesting to see how joint training and domain adaptation can influence L2BGAN to provide superior results on datasets like DarkFace.

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

APPENDIX

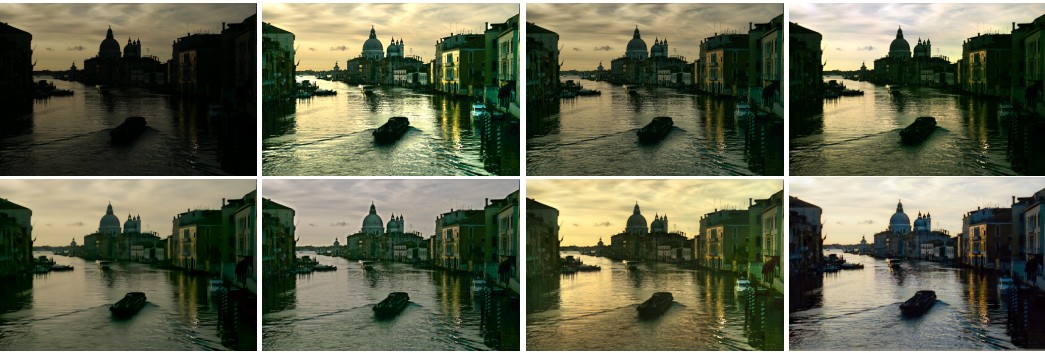

Figure 9: Sample original MEF image (top left) and its processed versions using Lime,FMSBIE,DeepUPE(first row, from second column to fourth column), and using MBLLEN, ZDCE, EGAN and L2BGAN (second row, from left to right).

## 5.1 EVALUATION ON RAW IMAGES

There is some research where image enhancement task is carried on raw data as input, which possess a lot of information not present in JPEG or PNG versions. Our aim is an algorithm for enhancing images whose raw version is not available: this is indeed the case for many applications where post-processing needs to be done on displayable formats. We have however performed a simple experiment where we have processed a raw image using our network, without any retraining. The results were better than the ones obtained using MBLLEN and can be found in fig10.

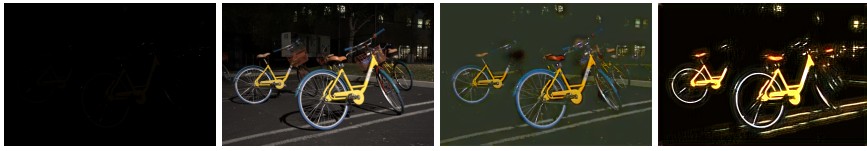

Figure 10: Original low light raw , GT, processed with L2BGAN and MBLLEN from left to right

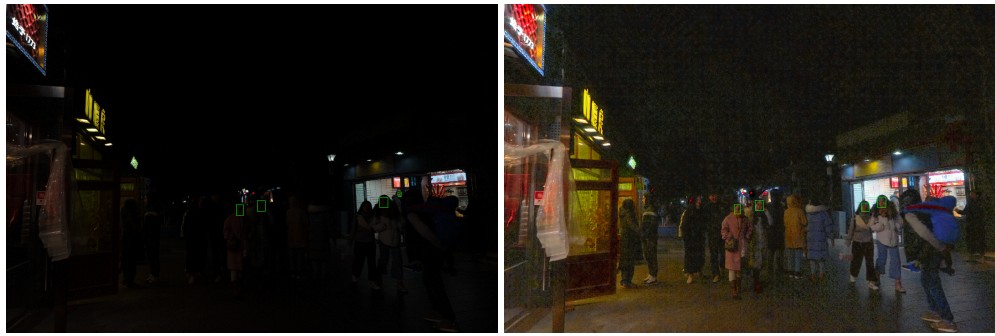

Figure 11: Face detection performances on original and preprocesse images from DARKFACE dataset. First column is original while the second column is the processed version using L2BGAN.

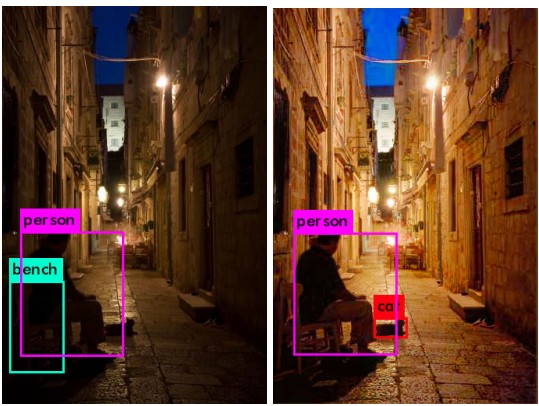

Figure 12: Face and object detection performances on original and preprocessed image from ExDark dataset. First column is original while the second column is the processed version using L2BGAN.

