# OpenReview forum: "L2BGAN: An image enhancement model for image quality improvement and image analysis tasks without paired supervision"
_ICLR.cc/2022/Conference — ICLR 2022 Submitted_

### Official Review · Reviewer_1V61 · 2021-10-21

**Correctness:** 3
**Technical Novelty And Significance:** 1
**Empirical Novelty And Significance:** 1
**Recommendation:** 3
**Confidence:** 4

**Main Review:**

Strengths:
This paper works on a CV task that is low-light image enhancement without paired data. It brings geometric and lighting prior to serving as assistant to the supervision.

Weaknesses:

This work focus on low light image enhancement. The title is improper and too broad.

The organization and writing of the paper are poor. For example, the introduction section has two paragraphs. The method only has one paragraph which mixes everything together.

The novelty of the paper is weak. Geometric and lighting priors are two frequently used priors for unsupervised low-light image enhancement. The constraints are based on simple priors. Specifically, it uses the image after rotation as the geometric prior and uses grammar corrected image as the lighting prior.

Some terms are used before explained, which makes them hard to understand.

The organization of Table 1 is also confusing.

In the experiments, more state-of-the-art methods should be compared.

An ablation study is necessary to show the effectiveness of the proposed priors and loss.

**Summary Of The Paper:**

This paper proposes a low light image enhancement method. It proposes using geometric and lighting consistency and a contextual loss.

**Summary Of The Review:**

This paper works on unsupervised low-light image enhancement. The novelty is weak. The organization and writing are poor. The validation needs improvement.

---

### Official Review · Reviewer_Fu5c · 2021-11-01

**Correctness:** 3
**Technical Novelty And Significance:** 2
**Empirical Novelty And Significance:** 2
**Recommendation:** 5
**Confidence:** 3

**Main Review:**

* Strengths:

Their model shows competitive performance for the benchmarks on DICM, LIME, MEF, NPE datasets. Their model outperforms previous models for the benchmarks on the LOL and BDD datasets.

Preprocessing the dart images by applying their model leads to better face detection performance compared to EnligthenGAN (Jiang et al., 2021). It also improves object detection performance somewhat, but it was not compared with previous works.

* Weaknesses:

Descriptions on their methodology in Section 3 is quite sketchy and confusing to follow. It is hard to get a clear picture of their formation on the full losses.

It is unclear what exactly their contextual loss is like. Eq. (3) is totally misleading if they adopted a contextual loss from "The contextual loss for image transformation with non-aligned data, R. Mechrez et al. ECCV 2018", which is never cited nor discussed clearly in the paper.

According to EnligthenGAN (Jiang et al., 2021) and UEGAN (Ni et al., 2020), the use of cycle-consistency is discouraged as it make the network more complex and harder to train. They need to address this issue and justify the use of cycle-consistency denied by recent similar works.

The ablation study is poorly made. They fail to convince the effectiveness of color, texture and edge discriminators, as well as geometric and lighting consistency.

References were not made properly: no citation on the contextual loss, no citation on the datasets used for their training and evaluation.

**Summary Of The Paper:**

This paper proposes an image enhancement model to translate a low light image to a bright image without a paired supervision through CycleGAN. Compared to similar recent GAN-based works such as EnligthenGAN (Jiang et al., 2021) and UEGAN (Ni et al., 2020), the proposed work relies on cycle-consistency, as well as geometric and illumination consistency. They also separate their discriminators for color, texture and edges.

**Summary Of The Review:**

The proposed model seems to lead some performance improvement in some benchmarks. However, the proposed methodology is not entirely convincing, requiring deeper analysis and justifications compared to recent GAN-based models. Also, it is hard to get a clear idea on their methodology in detail, as the paper is poorly written.

---

### Official Review · Reviewer_efio · 2021-11-02

**Correctness:** 2
**Technical Novelty And Significance:** 1
**Empirical Novelty And Significance:** 1
**Recommendation:** 3
**Confidence:** 5

**Main Review:**

1. The novelty in this paper is unclear. The paper mentions that "While none of such items are individually novel, exploiting their ensemble effect in this task is indeed a significant novel contribution." I am not sure if assembling the existing architecture and losses can be sufficient to warrant the publication of this paper in ICLR. Moreover, the results do not show any significant improvement over those of the existing methods.

2. It is well known in the community that unpaired GAN (or cycleGAN) can produce hallucination (fake contents). This important problem, particularly in image restoration/enhancement is not mentioned or addressed in this paper.

3. The paper mentions dealing with denoising and low-light enhancement problems simultaneously as the motivation. But why the method can deal with noise is unclear. More specifically, how the network balances the noise suppression and the over-smoothing effect is unclear.

4. The paper employs the grayscale image as the texture image. Why a simple grayscale operation can keep the texture information is unclear.  What is the benefit of using grayscale compared to rgb in low-light texture preservation?

5. These papers also use unpaired images:
* Yifan Jiang, Xinyu Gong, Ding Liu, Yu Cheng, Chen Fang, Xiaohui Shen, Jianchao Yang, Pan Zhou, and Zhangyang Wang. Enlightengan: Deep light enhancement without paired supervision. IEEE  Transactions on Image Processing, 30:2340–2349, 2021.
*Asha Anoosheh, Torsten Sattler, Radu Timofte, Marc Pollefeys, and Luc Van Gool. Night-to-day mage translation for retrieval-based localization. In 2019 International Conference on Robotics and Automation (ICRA), pp. 5958–5964. IEEE, 2019.
* Sharma et al., "Nighttime Stereo Depth Estimation using Joint Translation-Stereo Learning: Light Effects and Uninformative Regions." 2020 International Conference on 3D Vision (3DV). IEEE, 2020.
* Sharma et al., "Nighttime Visibility Enhancement by Increasing the Dynamic Range and Suppression of Light Effects", Computer Vision and Pattern Recognition, CVPR 2021.

**Summary Of The Paper:**

This paper proposes an image enhancement method using an unpaired-GAN based translation network (cycleGAN). It combines geometric, lighting consistency, and a contextual loss criterion. The geometric consistency is basically 90 rotations of the input image, while lighting consistency is basically gamma transformation.

**Summary Of The Review:**

The three important aspects of a good paper, i.e., contributions, novelty and performance, are missing in the paper.

---

### Official Review · Reviewer_Et8P · 2021-11-03

**Correctness:** 3
**Technical Novelty And Significance:** 2
**Empirical Novelty And Significance:** 3
**Recommendation:** 5
**Confidence:** 3

**Main Review:**

The main strength of the paper is represented by the results that it is able to obtain on standard datasets and how it compares to the state of the art.
On the downside, the main weakness of the paper is that none of the individual contributions is individually novel, e.g.:
- casting the low light enhancement problem as an unpaired image translation problem
- using geometric and illumination consistency
- using a contextual loss for semantic similarity
- using multiscale color, texture and edge discriminators
The paper can be therefore seen mainly as an incremental step in the field, and therefore not significant enough to permit its acceptance in this venue.



**Summary Of The Paper:**

The authors propose a new image enhancement specific for low light images. They exploit the concepts of geometric and lighting consistency together with a contextual loss criterion.
Extensive experiments are performed on benchmark datasets to compare their results both visually and objectively.

**Summary Of The Review:**

The work is incremental and uses already existing building blocks

---

### Decision · Program_Chairs · 2022-01-20

**Decision:**

Reject

**Comment:**

None of the reviewers championed the paper. Many weaknesses were shared across the reviewers: none of the individual contributions is individually novel, paper is not well written and the results do not show significant improvement over the prior state of the art. No rebuttal was provided. The AC agrees with the reviewers that the paper is not ready for publication at ILCR.